# 3D-Printed GelMA/PEGDA/F127DA Scaffolds for Bone Regeneration

**DOI:** 10.3390/jfb14020096

**Published:** 2023-02-09

**Authors:** Jianpeng Gao, Ming Li, Junyao Cheng, Xiao Liu, Zhongyang Liu, Jianheng Liu, Peifu Tang

**Affiliations:** 1Department of Orthopaedics, Chinese PLA General Hospital, Beijing 100039, China; 2Medical School of Chinese PLA, Beijing 100039, China; 3National Clinical Research Center for Orthopedics, Sports Medicine & Rehabilitation, Beijing 100853, China

**Keywords:** biomaterials, bone regeneration, bone tissue engineering, digital light processing

## Abstract

Tissue-engineered scaffolds are an effective method for the treatment of bone defects, and their structure and function are essential for bone regeneration. Digital light processing (DLP) printing technology has been widely used in bone tissue engineering (BTE) due to its high printing resolution and gentle printing process. As commonly used bioinks, synthetic polymers such as polyethylene glycol diacrylate (PEGDA) and Pluronic F127 diacrylate (F127DA) have satisfactory printability and mechanical properties but usually lack sufficient adhesion to cells and tissues. Here, a compound BTE scaffold based on PEGDA, F127DA, and gelatin methacrylate (GelMA) was successfully prepared using DLP printing technology. The scaffold not only facilitated the adhesion and proliferation of cells, but also effectively promoted the osteogenic differentiation of mesenchymal stem cells in an osteoinductive environment. Moreover, the bone tissue volume/total tissue volume (BV/TV) of the GelMA/PEGDA/F127DA (GPF) scaffold in vivo was 49.75 ± 8.50%, higher than the value of 37.10 ± 7.27% for the PEGDA/F127DA (PF) scaffold and 20.43 ± 2.08% for the blank group. Therefore, the GPF scaffold prepared using DLP printing technology provides a new approach to the treatment of bone defects.

## 1. Introduction

Bone defects caused by tumors, infections, and trauma are difficult and critical to treat clinically and thus require effective therapeutic initiatives [1,2,3]. Bone tissue engineering (BTE) scaffolds have been widely used in bone repair because they are capable of filling the defective area, providing mechanical support, and guiding the growth of new tissues in the early treatment of bone defects [4,5,6].

With recent developments in 3D printing technology, it has become possible to prepare scaffolds according to a predesigned (computer aided design, CAD) structure and achieve more precise control over the macroscopic structure of the scaffold [7,8,9]. Digital light processing (DLP) technology is a representative lithography-based 3D bioprinting technology characterized by a layer-by-layer based printing pattern, the core of which is the digital micromirror device (DMD), which provides superior image stability, fidelity, and reliability. Visible or ultraviolet light can be used to cross-link bioinks and complete the liquid–solid conversion in DLP printing [10,11,12,13,14]. Thus, some traditional natural and synthetic materials, which previously could only be printed by extrusion printers, are now available for printing via DLP, a gentler printing method, as long as the bioinks can be endowed with light-curing properties (e.g., Gel-GelMA, PEG-PEGDA, and F127-F127DA) [15,16,17,18,19].

Polyethylene glycol (PEG), which exhibits high biocompatibility and almost no immunogenicity, can be chemically modified to form polyethylene glycol diacrylate (PEGDA), which possesses photo-crosslinking properties, low viscosity, and high solubility, making it an ideal biomaterial for DLP bioprinting [18,20,21]. Despite its many advantages, PEGDA is generally inelastic and brittle, which makes it more likely to be used in combination with other materials rather than alone for bone tissue engineering [22]. It has been shown that PEGDA can be mixed with materials such as nanohydroxyapatite [23,24,25], nanoclay [26], and extracellular matrix [27] to form bioinks for the preparation of BTE scaffolds, which play an important role in the treatment of bone defects. However, the mechanical properties, printability, and bioactivity of these scaffolds still need to be improved. Pluronic F127 diacrylate (F127DA), modified with Pluronic F127 (F127), exhibits low swelling properties, fatigue resistance, and proper elastic modulus, which complement the shortcomings of PEGDA [28,29,30]. Bioinks combined with F127DA and PEGDA may possess excellent printability and mechanical properties. However, this combination has not been widely used in BTE perhaps due to the lack of cell adhesion [31,32,33].

Gelatin methacrylate (GelMA), as one of the most commonly used photosensitive hydrogel materials in bone tissue engineering, has good biocompatibility [34]. Unlike PEGDA and F127DA, GelMA can significantly promote cell adhesion and proliferation [35]. Thus, based on this, a new GelMA/PEGDA/F127DA bioink was developed in this study and a GelMA/PEGDA/F127DA scaffold was prepared using DLP printing technology; the scaffold not only possessed good mechanical properties similar to synthetic materials and played a supporting role in early implantation, but the porous structure created by 3D printing also actively promoted the growth of bone tissue. Moreover, the addition of GelMA greatly increased the cell adhesion of the scaffold, which is crucial for osteoconduction and bone regeneration. Therefore, the composite scaffold consisting of natural (GelMA) and synthetic materials (PEGDA-F127DA) prepared using DLP technology represents a promising approach for the treatment of bone defects (Figure 1).

## 2. Materials and Methods

### 2.1. Materials

Gelatin from porcine skin was purchased from Sigma-Aldrich (St. Louis, MO, USA). Methacrylic anhydride (MA, 97%) and lithium phenyl-2,4,6-trimethylbenzoyl phosphinate (LAP) were purchased from J&K (Beijing, China). PEGDA and F127DA were purchased from Engineering for Life (EFL) (Suzhou, China). Cell counting kit-8 (CCK-8), a live/dead viability assay kit, phalloidin, Alizarin Red S, and an alkaline phosphatase (ALP) assay kit were purchased from Beyotime (Shanghai, China).

### 2.2. Preparation of Bioinks

The preparation of GelMA was carried out as described previously [36]. In brief, gelatin was dissolved in phosphate buffered saline (PBS) at 40 °C to prepare a 10% gelatin solution. After adding methacrylic anhydride (MA) dropwise into the gelatin solution, the solution was stirred with magnetic force for three hours at 40 °C and 300 rpm; then, a white porous foam was prepared after dialysis against distilled water for 5 days at 40 °C and lyophilized.

GelMA (5% (*w*/*v*)), PEGDA (10% (*w*/*v*)), and F127DA (5% (*w*/*v*)) were dissolved in PBS supplemented with lithium phenyl (2,4,6-trimethylbenzoyl) phosphinate (LAP, 0.25% (*w*/*v*)) and tartrazine (0.05% (*w*/*v*)). Then, a bioink composed of 5% GelMA/10% PEGDA/5% F127DA (GPF) was prepared. The preparation method of another bioink containing 10% PEGDA/5% F127DA (PF) was the same.

### 2.3. Fabrication of Scaffolds

The CAD model was designed as a cylinder with interconnected pores, with a diameter of 6 mm, height of 8 mm, and pore size of 600 μm. Then, a DLP printer (BP8601 Pro, EFL, Suzhou, China) was used to prepare the scaffolds and the parameters were adjusted for printing. Then, the scaffolds were strengthened under ultraviolet light for 3 min (kernel parameters: layer height, 100 μm; light intensity, 20 mW/cm^2^; exposure time, 4 s; temperature, 29 °C).

### 2.4. Characterization

#### 2.4.1. Microstructure of the 3D-Printed Scaffolds

The scaffolds were observed using a scanning electron microscope (SEM, SU8100, HITACHI, Hitachi, Japan) after lyophilization (K850, Quorum, East Sussex, UK) and gold/palladium sputter-coating (MC1000, HITACHI, Hitachi, Japan). The pore size of the printed scaffolds was calculated using the ImageJ software (V1.8.0, NIH, Bethesda, MD, USA); three images were selected for each sample, and five pores were measured for each image.

#### 2.4.2. Compressive Tests

A compression test was performed using a universal tensile machine (3365, Instron, Boston, MA, USA) at room temperature. The compression modulus was defined as the initial slope of the linear region of the stress–strain curve. The mechanical indexes (compressive stress and modulus) were acquired according to the software (n = 3).

#### 2.4.3. Swelling

The different scaffolds were placed into PBS and soaked for 24 h at 37 °C, and their weights (Ws) were measured after sufficient swelling. Then, the scaffolds were freeze-dried to obtain their dry weight (Wd). The swelling ratio was calculated as
(1)$$\mathrm{Swelling}\;\mathrm{ratio}=\frac{\mathrm{Ws}-\mathrm{Wd}}{\mathrm{Wd}}$$

#### 2.4.4. Degradation

The different scaffolds were lyophilized, and their weights (W0) were measured. Then, the lyophilized scaffolds were placed in PBS solution and soaked at 37 °C. The PBS was changed every two days and the samples were removed on the 3rd, 6th, 9th, 12th, 15th, 20th, 25th, 30th, 40th, and 50th day. After rinsing twice with deionized water, the samples were lyophilized and weighed (Wt). The remaining weight was calculated as
(2)$$\mathrm{Remaining}\;\mathrm{weight}\;\left(\%\right)=\frac{\mathrm{Wt}}{W0}\times 100\%$$

### 2.5. Cell Culture

The mouse embryo osteoblast precursor cells (MC3T3-E1 subclone 14) and rabbit bone marrow mesenchymal stem cells (rBMSCs) used in this experiment were obtained from the Orthopedic Laboratory of the PLA General Hospital. The original generation of cells was expanded to the 3rd generation with medium containing α-MEM, fetal bovine serum (FBS, 10%) and penicillin–streptomycin (1%) for experiments. Both types of cells were cultured in this medium—refreshed every 2 days—in a 37 °C and 5% CO_2_ environment.

### 2.6. Cell Viability

#### 2.6.1. Extracts of Different Scaffolds

The scaffolds were soaked in medium for 48 h at 37 °C. For different scaffolds, 100% extracts were prepared according to the standard of 1.25 cm^2^/mL and diluted to different concentrations of 75%, 50%, and 25%.

#### 2.6.2. CCK-8

MC3T3-E1 cells were cultured in extracts with different concentrations of different scaffolds for 1–5 days. After the color deepened for 1 h with the addition of CCK-8 (10%), the cell viability was analyzed using a microplate reader (Thermo Fisher, Waltham, MA, USA).

#### 2.6.3. Live/Dead Staining of Cells Cultured with Extracts

MC3T3-E1 cells were cultured for 48 h in the extracts at the optimal concentration obtained via the CCK-8 assay. The cells were incubated with live/dead dye for 15 min, and then observed under a fluorescence microscope (Ni-U, Nikon, Tokyo, Japan), where green represented living cells and red represented dead cells.

#### 2.6.4. Phalloidin Staining of Cells Cultured with Extracts

MC3T3-E1 cells cultured for 48 h in the extracts were fixed in 4% paraformaldehyde solution for 30 min. After three washes with PBS, the cells were stained with phalloidin for 30 min and DAPI for 5 min. The morphology of the cytoskeleton was observed with a confocal microscope (FV3000, Olympus, Tokyo, Japan).

#### 2.6.5. Live/Dead Staining of Cells Cultured on Scaffolds

Sterile scaffolds were soaked in medium for 15 min in 24-well plates; following their removal from the medium, 1 mL of cell suspension (5 × 10^4^ cells) was added. MC3T3-E1 cells were seeded on different scaffolds and cultured in medium for 48 h. The cells were incubated with live/dead dye for 15 min, and then observed under a confocal microscope (Olympus, FV3000, Tokyo, Japan), where green represented living cells and red represented dead cells.

### 2.7. Effect of the Scaffold on Osteogenic Differentiation In Vitro

#### 2.7.1. Extracts of Different Scaffolds

To distinguish the promotive effect of the scaffolds on osteogenesis under osteoinductive conditions (OIC) and non-osteoinductive conditions (non-OIC), osteoinductive extracts were prepared in a similar way to the normal extracts.

rBMSCs cultured for 5 (non-OIC) and 7 (OIC) days were fixed in 4% paraformaldehyde solution for 30 min. After three washes with PBS, they were stained with an ALP assay kit for one hour and observed with a stereomicroscope (SMZ25, Nikon, Tokyo, Japan).

#### 2.7.2. Alizarin Red S

rBMSCs cultured for 5 (non-OIC) and 21 (OIC) days were fixed in 4% paraformaldehyde solution for 30 min. After three washes with PBS, they were stained with Alizarin Red S for ten minutes and observed with a stereomicroscope (SMZ25, Nikon, Tokyo, Japan).

#### 2.7.3. Quantitative Real-Time PCR

The scaffolds were soaked in medium for half an hour and then removed. A total of 10^5^ cells were seeded on the surface of the scaffolds and cultured for 24 h in normal medium, which was then replaced with osteoinductive medium (50 µg/mL ascorbic acid, 10 mM β-phosphoglycerol, and 10 nM dexamethasone) or osteoinductive extracts. After 5 days of culture, total RNA was extracted from the cells for real-time PCR using Trizol reagent (G3013, Servicebio, Wuhan, China); each sample was repeated three times.

### 2.8. Effect of the Scaffold on Bone Regeneration In Vivo

#### 2.8.1. Ethics Statement

All animals used in this study were obtained from the Animal Experiment Center of the PLA General Hospital and approved by the Ethics Committee (2022-x18-51).

#### 2.8.2. Implantation in Rabbit Femoral Condyle Defects

New Zealand white rabbits (2.5 kg ± 0.5 kg, male, 3 in each group) were used in this experiment. Briefly, the rabbits were anesthetized, and the distal femur was shaved and disinfected. After cutting the skin and subcutaneous tissue, a cylindrical defect 6 mm in diameter was created, without penetrating the contralateral cortex in the distal femur, using a surgical drill. The sterile scaffolds were inserted into the defect site and the subcutaneous tissue and skin were sutured layer by layer. The rabbits were sacrificed at week 4 and week 12 postoperatively for the next step of treatment.

#### 2.8.3. Micro-CT Analysis

The Inveron MM System (Siemens, Munich, Germany) was used to evaluate the amount of new bone in each group of rabbits via micro-CT scans. The scanning parameters were an effective pixel size of 17.34 μm, a current of 500 μA, a voltage of 80 kV, and an exposure time of 1500 ms. The 2D images were reconstructed into 3D images using Inveron Research Workplace (Siemens) to calculate the bone regeneration parameters: BMD, bone volume/total volume (BV/TV), trabecular thickness (Tb.Th), and trabecular spacing (Tb.SP).

#### 2.8.4. Histology Analysis

Samples were decalcified in 10% EDTA, dehydrated in a stepped concentration of ethanol, and cleared using xylene. The samples were then embedded in paraffin and cut into 10 mm slices using a microtome for staining.

### 2.9. Statistical Analysis

The results between two groups were analyzed using a paired t test. The results among three groups were analyzed using a one-way analysis of variance (ANOVA) with a Tukey–Kramer multiple comparison analysis using the GraphPad Prism software (version 8, GraphPad, San Diego, CA, USA). The data are expressed as the mean ± standard deviation (SD) and all experiments were performed at least three times. A value of *p* < 0.05 was regarded as statistically significant (* *p* < 0.05, ** *p* < 0.01, *** *p* < 0.001).

## 3. Results

### 3.1. Characterization

Two scaffolds with different compositions, 10% PEGDA/5% F127DA (PF) and 5% GelMA/10% PEGDA/5% F127DA (GPF), were successfully prepared. As predicted, the printed scaffolds possessed a favorable porous structure, with a pore size of 508.13 ± 21.28 μm for the GPF scaffold, which was lower than the 564.04 ± 17.56 μm found for the PF scaffold and 600 μm for the CAD (Figure 2A–E). However, as shown in Figure 2H–J, the addition of GelMA did not significantly change the compressive strength of the scaffolds, but the modulus decreased from 127.4 ± 12 kPa to 92.34 ± 6.80 kPa as the scaffold became more elastic. In addition, when the scaffolds were soaked in PBS for 24 h to reach swelling equilibrium, the swelling ratios of PF and GPF were 3.35 ± 0.88 and 3.90 ± 0.62, respectively. The remaining weight of GPF at day 50 was 57.81 ± 3.64%, which may be more suitable for bone regeneration than the weight of 67.10 ± 4.30% observed for PF (Figure 2F,G).

### 3.2. Biocompatibility

Good biocompatibility is the basis for the clinical application of bone tissue engineering. To investigate the cytocompatibility of PF and GPF, the extracts were prepared at different concentrations (100%, 75%, 50%, and 25%) for cell culture according to the standards of extract preparation. As shown in Figure 3A, there was no obvious effect of either scaffold on cell proliferation when the extract concentrations were 100%,75%, or 50%, while GPF significantly improved cell proliferation after day 4 compared to PF when the extract concentration was 25%. Moreover, cells were cultured in the 25% extract for 48 h and stained with a live/dead viability assay kit, and phalloidin, and similar cell numbers and morphologies were observed for PF, GPF, and normal medium (Figure 3B,C and Figure 4).

For further observation of the growth condition of the cells on the scaffold surface, MC3T3-E1 cells were inoculated on the scaffolds and cultured for 48 h. After staining with a live/dead viability assay kit, it was found that the number of cells on the surface of the PF scaffolds was low and most were dead, and the cells exhibited a spherical shape. In contrast, cells on the surface of the GPF scaffolds were observed to be well proliferated, with a low number of dead cells; in addition, MC3T3-E1 cells could extend their tentacles on the surface of the GPF scaffolds (Figure 5).

### 3.3. Capacity for Osteogenic Differentiation In Vitro

To distinguish the osteogenic promotion of the scaffolds under OIC and non-OIC, we performed an in vitro validation of osteogenic differentiation in each of the two conditions.

The rBMSCs were stained with an ALP assay kit and Alizarin Red S after 5 days of culture in normal extracts. The osteogenic differentiation of rBMSCs was not promoted by either the PF or GPF scaffolds (Figure 6A,B), and the expression of osteogenic genes such as Col-1, OPN, OCN, and Runx2 was similar in each group after seeding the cells on the scaffolds for 5 days (Figure 6C).

However, when rBMSCs were cultured in osteogenic extracts, a clear difference was observed between ALP staining on day 7 and Alizarin Red S staining on day 14, where GPF possessed a greater ability to promote osteogenic differentiation than PF (Figure 7A,B). Similarly, the expression of osteogenic-related genes was higher in cells seeded on GPF rather than PF scaffolds (Figure 7C).

### 3.4. Bone Regeneration In Vivo

To explore the effect of the different scaffolds on the treatment of bone defects, PF and GPF scaffolds were implanted at the distal femoral defect site and a micro-CT was performed at week 4 and week 12 postoperatively. The results suggested that without intervention, there was only a small amount of new bone at the defect site at week 12. However, greater regeneration of bone tissue was observed with both the PF and GPF scaffolds, mostly from cancellous bone toward cortical bone, and the new bone took on a scaffold-like meshed shape. At week 12, the new bone latticed off, which could be related to degradation inside the scaffold (Figure 8A). Similar results to the CT images can be observed in Figure 8B, with the GPF scaffold achieving better efficacy in BMD, BV/TV, Th. Tb, and Th. Sp.

Subsequently, a histological analysis of the samples was performed at week 12. It was observed from HE and Masson staining that a gap existed between the PF scaffolds and the new bone organization, which was consistent with the in vitro study where the PF scaffolds lacked cell and tissue adhesion. In contrast, the new bone could adhere to the surface of the GPF scaffold and grow into the pores, exhibiting good osteoconductivity (Figure 9).

## 4. Discussion

DLP printing technology is widely used in bone tissue engineering owing to its good printing accuracy and gentle printing process, which requires bioink with photosensitive properties [37,38,39]. At this stage, the materials used mainly consist of natural materials (GelMA, HAMA, etc.), synthetic materials (PEGDA, F127DA, PPF, etc.), and inorganic materials (TCP, HA, metals, etc.). Among them, natural and synthetic materials can be chemically modified to endow them with photosensitive properties for direct DLP printing [40,41], while inorganic materials need to be mixed into a photosensitive resin for printing and sintered (1200 °C) to remove organics [42]. Greeshma et al., prepared GelMA-based bioink from autologous bone particles (BPs) and determined the appropriate printing parameters, revealing that 3D-printed GelMA/BP-based composite scaffolds could effectively promote bone regeneration by improving the proliferation, migration, and osteogenic differentiation capacity of cells [16]. Zhang et al., prepared Haversian bone-mimicking scaffolds using the DLP printing technique with bioceramics; the scaffolds were found to induce osteogenesis, angiogenesis, and neural differentiation in vitro and accelerate the growth of blood vessels and new bone formation in vivo [43].

Natural materials with good bioactivity have some disadvantages such as poor mechanical properties and rapid degradation [44]. Inorganic ceramic and some metallic materials (e.g., titanium, steel, etc.) with strong mechanical properties possess slow degradation rates, and are generally brittle [45]. In addition, some metals with weak rigidity, such as magnesium, can promote vascular and bone regeneration; however, some studies have demonstrated that the dynamic in vivo environment can lead to the accelerated fatigue of magnesium materials, making the magnesium scaffold lose its abilities as described earlier [44]. In addition, high-temperature sintering during the preparation of scaffolds using inorganic materials can lead to the inactivation of active substances in bioink; these disadvantages limit the application of these materials in BTE. Therefore, synthetic polymers need to be further explored for use in DLP-printed bone tissue engineering. PEGDA (brittle material) and F127DA (elastic material) are the most common synthetic polymers used for modification, possessing photo-crosslinking and complementary mechanical properties. Shen et al., prepared a tissue adhesive with good histocompatibility using PEGDA and F127DA; the adhesive was expected to repair wounded tissues without suturing [45]. However, neither material has been proposed for the preparation of BTE scaffolds via DLP.

In this study, PEGDA and F127DA were added to bioink and showed satisfactory printing and mechanical properties, while no obvious cytotoxicity of the printed scaffolds was observed. However, cells exhibited difficulty in adhering after seeding on the surface of the PF scaffold, which may be a hindrance for its application in BTE. Therefore, the addition of materials with stronger adhesion properties is needed to increase the bioactivity of synthetic polymers. Wang et al., prepared an injectable hydrogel by adding GelMA to a PEGDA-based bioink. Although the compressive strength of the scaffold (approximately 300 kPa) still needs to be improved and its injectable properties imply the abandonment of the macroscopic porous structure, the addition of GelMA improved the bioadhesion of the bioink [25,46]. Therefore, GelMA, with its good biocompatibility and adhesion support, was added to the PEGDA/F127DA bioink in this study to promote cell adhesion and proliferation. It can be clearly observed in Figure 5 that cells on the surface of the GPF scaffolds can extend their tentacles for better adhesion and proliferation than the spherical cells on the surface of the PF scaffolds. In addition, while retaining the good mechanical properties of the PF scaffold, the GPF scaffold had a faster degradation rate, which could help new bone to better replace the material, facilitating the regeneration of bone tissue.

In addition to the choice of material, pore size is also crucial for cell proliferation and differentiation. From the literature, pore sizes larger than 300 μm show better vascularization and osseointegration in BTE [47,48]. Zhang et al., showed that the optimal pore size for osteogenic capacity is approximately 600–700 μm, and pore sizes that are too small or too large affect cell behavior and bone regeneration [49]. Chen et al., demonstrated that a pore size of 500 μm showed the best cell proliferation and differentiation and inward bone growth [50]. Luo et al., concluded that porous scaffolds with a pore size of 400–600 μm better promote osteogenesis and osseointegration [51]. Although some scholars believe that a small pore size (188 μm) is more favorable for the osteogenic differentiation of cells in vitro [52], more studies have demonstrated that 400–700 μm is a good choice for the preparation of bone tissue engineering scaffolds [49,50,51,53]. In this study, the standard pore size of the scaffold was 600 μm, while the printed PF scaffold had a pore size of approximately 564 μm and the pore size of the GPF scaffold was approximately 508 μm. The addition of GelMA increased the bioadhesion of the scaffold but reduced the printability of the bioink; nevertheless, both scaffolds had a good pore size structure and exhibited satisfactory bone regeneration.

The conditions under which GPF promotes bone regeneration were also investigated in this study. There are various ways for materials to enhance bone regeneration, one of which is to induce the osteogenic differentiation of rBMSCs when there is no exogenous induction; another is to use a scaffold to accelerate the osteogenic differentiation of rBMSCs when an inductive environment exists. The GPF scaffolds in this study represent the second option, which can be observed from the in vitro experiments. When rBMSCs were cultured in normal medium, the GPF scaffolds and PF scaffolds did not exhibit osteogenesis-promoting effects, whereas when cultured in osteogenesis-inducing medium, the effect of the GPF scaffolds in promoting osteogenesis was significantly greater than that of the PF scaffolds. Combined with the 3D images of the defect site from the CT reconstruction, these results indicated that GPF can play a facilitating role in promoting bone regeneration when an osteogenesis-inducing environment exists in vivo. In addition, the lattice-like new bone showed that the bone was growing and crawling along the pore structure. At 12 weeks, the reduction of the lattice-like structure indicated the degradation of the scaffold, leading to the loss of its original aperture which was replaced by new bone tissue. Our histological observations, shown in Figure 9, were consistent with the in vitro structure. A lack of adhesion of the PF scaffold to the cells and tissues resulted in a significant gap between them in vivo. In contrast, the tightly adherent growth of bone tissue could be observed around the GPF scaffold containing GelMA, indicating that the GPF scaffold could guide the adhesion and regeneration of new bone.

Our results clearly indicate that GPF possesses a satisfactory porous structure (508.13 ± 21.28 μm) and mechanical properties (829.59 ± 89.21 kPa) to promote osteogenic differentiation under osteoinductive conditions and guide bone growth in vivo. However, the following limitations may exist. First, the accuracy of the universal tensile machine is 0.5%, which means the mechanical results may have an error of 0.5%. Second, the differential effects of GPF scaffolds on osteogenic differentiation in diverse environments still require further investigation. Third, the establishment of a 3D finite element model to simulate 3D physiological loading has made significant progress in the design of implants [54], which has given us great insight to predict the state of BTE scaffolds in vivo through computational simulations for developing an optimal structure.

## 5. Conclusions

GPF scaffolds prepared using DLP printing technology not only possess satisfactory mechanical properties, but also an appropriate degradation rate that is more compatible with the time course of bone regeneration. By improving the disadvantages of traditional synthetic polymers that are not conducive to cell adhesion, these scaffolds exhibit excellent histocompatibility, guiding the new bone tissue to grow inside the defect when implanted in vivo. Furthermore, GPF scaffolds can effectively promote the osteogenic differentiation of rBMSCs in an osteoinductive but not a non-osteoinductive environment. The reasons for such different results will be the focus of further research. In summary, GPF-based composite scaffolds prepared using DLP printing technology provide a new approach to the clinical treatment of bone defects.

## Figures and Tables

**Figure 1 jfb-14-00096-f001:**
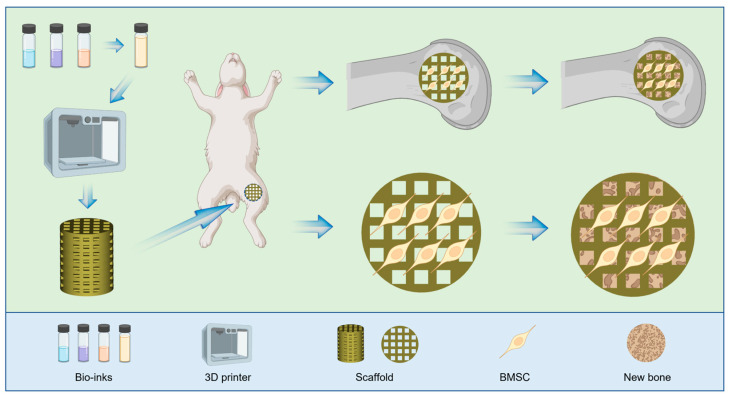
Schematic of the GelMA/PEGDA/F127DA scaffold for bone regeneration.

**Figure 2 jfb-14-00096-f002:**
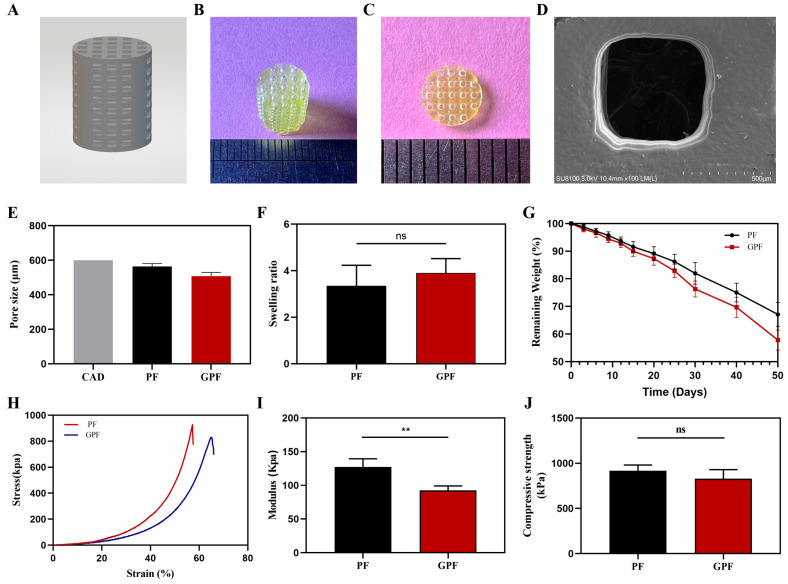
Characterization of scaffolds. (**A**) Image of 3D modeling. (**B**,**C**) Images of the GPF scaffold. (**D**) SEM image of the GPF scaffold. (**E**) Pore size of the CAD and scaffolds. (**F**) Swelling ratio of different scaffolds. (**G**) In vitro degradation behavior of the two scaffolds in PBS (37 °C, pH = 7.4). (**H**) Compressive stress–strain curves of the scaffolds. (**I**) Compressive modulus of the scaffolds. (**J**) Compressive strength of the scaffolds. Data were analyzed using a paired t test and are shown as the mean ± standard deviation (** *p* < 0.01, n = 3).

**Figure 3 jfb-14-00096-f003:**
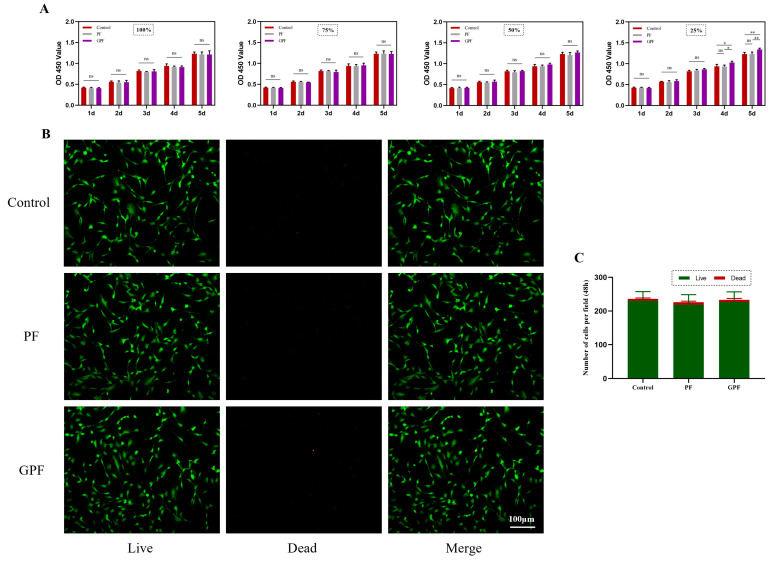
Biocompatibility of scaffolds as extracts in vitro. (**A**) CCK-8 assay showing the proliferation of MC3T3-E1 cells co-cultured with different extracts of different scaffolds for 1–5 days. (**B**) Live/dead assay of MC3T3-E1 cells co-cultured with 25% extract for 2 days. Green represents living cells and red represents dead cells. (**C**) Number of cells in the live/dead assay. Data were analyzed via a one-way ANOVA and are shown as the mean ± standard deviation (* *p* < 0.05, ** *p* < 0.01, n = 3).

**Figure 4 jfb-14-00096-f004:**
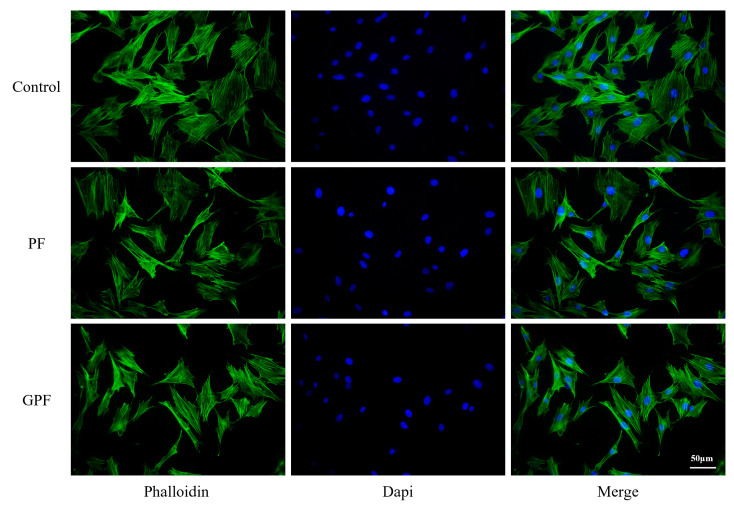
Phalloidin staining of MC3T3-E1 cells cocultured with 25% extract for 2 days.

**Figure 5 jfb-14-00096-f005:**
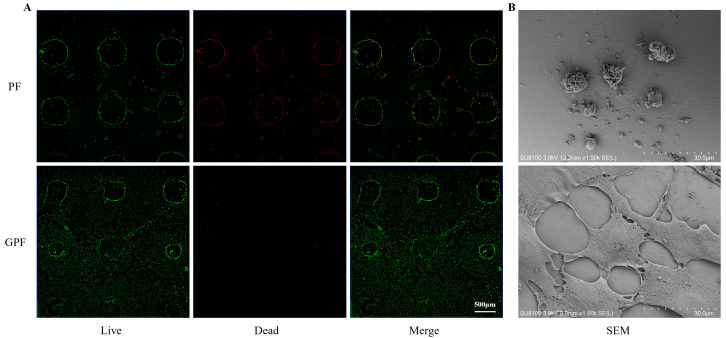
Biocompatibility and SEM images of scaffolds in vitro. (**A**) Live/dead assay of MC3T3-E1 cells cultured on scaffolds for 2 days. Green represents living cells and red represents dead cells. (**B**) SEM images of cells cultured on the surface of the scaffolds for 2 days.

**Figure 6 jfb-14-00096-f006:**
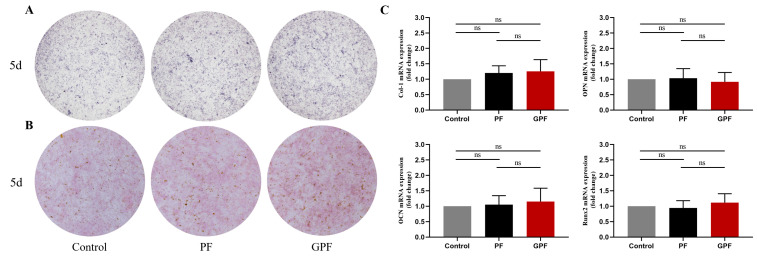
Effect of scaffolds on osteogenic differentiation under non-osteoinductive conditions. (**A**) ALP staining after 5 days of culture in normal extracts. (**B**) Alizarin Red S staining after 5 days of culture in normal extracts. (**C**) Expression of osteogenic-related genes determined using quantitative real-time PCR. Data were analyzed via a one-way ANOVA and are shown as the mean ± standard deviation (n = 3).

**Figure 7 jfb-14-00096-f007:**
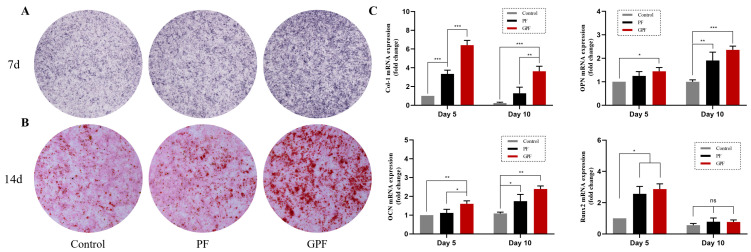
Effect of scaffolds on osteogenic differentiation under osteoinductive conditions. (**A**) ALP staining after 7 days of culture in osteoinductive extracts. (**B**) Alizarin Red S staining after 14 days of culture in osteoinductive extracts. (**C**) Expression of osteogenic-related genes determined using quantitative real-time PCR. Data were analyzed via a one-way ANOVA and are shown as the mean ± standard deviation (* *p* < 0.05, ** *p* < 0.01, *** *p* < 0.001, n = 3).

**Figure 8 jfb-14-00096-f008:**
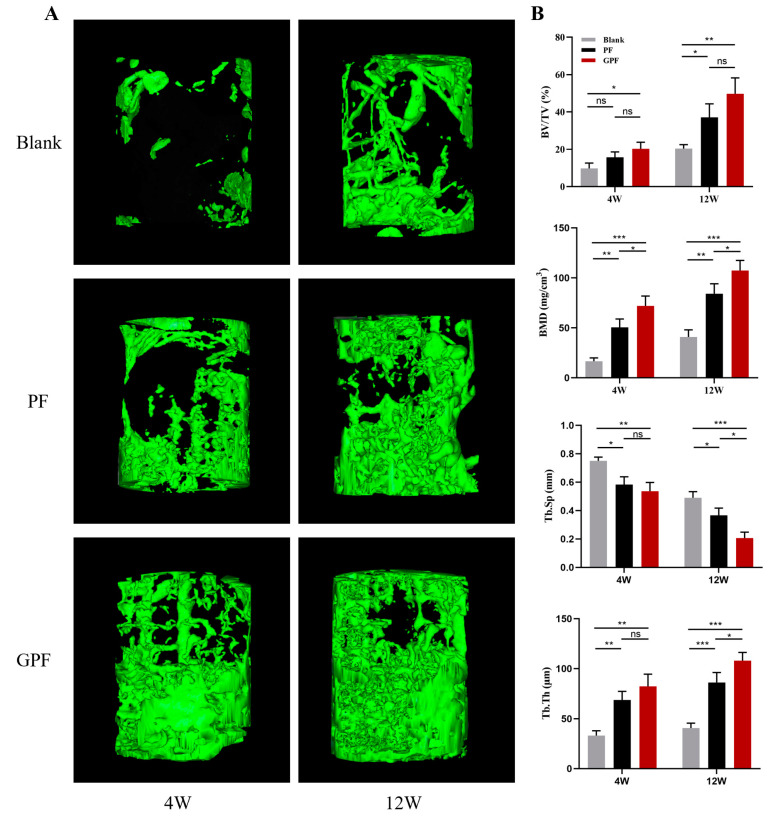
Micro-CT analysis of new bone formation at 4 weeks and 12 weeks. (**A**) Reconstructed 3D patterns from micro-CT images of femur defects at 4 weeks and 12 weeks. (**B**) Micro-architectural parameters of the newly formed bone. BMD-bone mineral density; BV/TV-bone tissue volume/total tissue volume; Tb.Th-trabecular thickness; Tb.Sp-trabecular separation. Data were analyzed via a one-way ANOVA and are shown as the mean ± standard deviation (* *p* < 0.05, ** *p* < 0.01, *** *p* < 0.001, n = 3).

**Figure 9 jfb-14-00096-f009:**
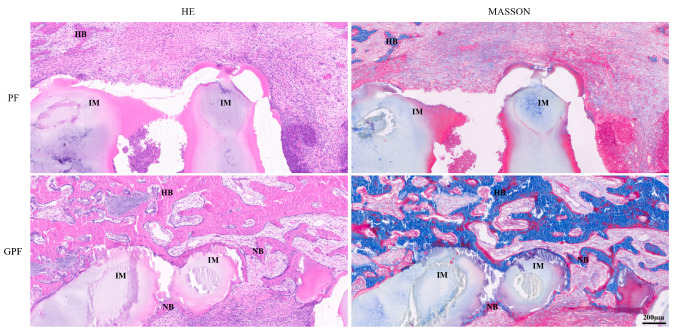
Histological evaluation of newly formed bone at 12 weeks. NB-new bone; HB-host bone; IM-implanted materials.

## Data Availability

The data supporting the reported results can be provided by the authors on request.

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
