# Peer review of "3D-Printed GelMA/PEGDA/F127DA Scaffolds for Bone Regeneration"

_jfb, 2023, doi:10.3390/jfb14020096_

Round 1
Reviewer 1 Report
In this paper, the authors have evaluated the in vitro and in vivo effect of scaffolds with composite composition prepared by DLP printing technology on bone regeneration. The article is interesting. Thus, I suggest a minor revision for the improvement thereof.
For that, I suggest the follows issues:
Materials and methods:
1) In the sentence: "The preparation of GelMA was as described previously". The author should add a bibliographic reference for that.
2) the authors used MC3T3-E1 cells in the experiments. Which subclone were used?
3) the authors should describe better how the cells were incorporated inside of the scaffolds or how they seeded the cells inside of the scaffold.
4) What was the sample size (n=?) used in the in vivo experiment? Please, add in the manuscript.
5) Which statistical test was used to compare the results of figure 1B, 1E, and 1F? T-test? Please, add this information on the statistical analysis section.
Author Response
Dear Reviewer,
Thank you for your insightful advice on our manuscript entitled “DLP-Printed Scaffold Based on Natural-synthetic Polymers for Bone Regeneration” (ID: jfb-2158500). Accordingly, we have revised the manuscript and all amendments are marked up using the “Track Changes” function. In addition, point-by-point responses are listed below this letter.
We hope that the revision is acceptable for publication.
Yours sincerely,
Jianheng Liu
January 25, 2023
Comments:
- In the sentence: "The preparation of GelMA was as described previously". The author should add a bibliographic reference for that.
Response: Thank you for the kind reminders. We have added the reference to the manuscript in the right place. - the authors used MC3T3-E1 cells in the experiments. Which subclones were used?
Response: Thank you for your insightful suggestions. The MC3T3-E1 cells we used in the experiments are subclone 14. We have added the details to the Materials and Methods section in line 120. - the authors should describe better how the cells were incorporated inside of the scaffolds or how they seeded the cells inside of the scaffold.
Response: Thank you for your comments. Sterile scaffolds were soaked in medium for 15 min in 24-well plates, which were then removed from the medium, and 1 ml of cell suspension (5 × 104 cells) was added. MC3T3-E1 cells were seeded on different scaffolds and cultured in medium for 48 h. We have added the details to the Materials and Methods section in line 147. - What was the sample size (n=?) used in the in vivo experiment? Please, add in the manuscript.
Response: Thank you for your comments. The sample size used in the in vivo experiment was three. We have added the details to the Materials and Methods section in line 178. - Which statistical test was used to compare the results of figure 1B, 1E, and 1F? T test? Please, add this information on the statistical analysis section.
Response: Thank you for your comments. The data were analyzed by paired t tests and are shown as the mean ± standard deviation. We have added the details in the figure legend in line 222.
Reviewer 2 Report
Please see the comments in the attached PDF.

Author Response
Dear Reviewer,
Thank you for your insightful advice on our manuscript entitled “DLP-Printed Scaffold Based on Natural-synthetic Polymers for Bone Regeneration” (ID: jfb-2158500). Accordingly, we have revised the manuscript and all amendments are marked up using the “Track Changes” function. In addition, point-by-point responses are listed below this letter.
We hope that the revision is acceptable for publication.
Yours sincerely,
Jianheng Liu
January 25, 2023
Comments:
- The title of the paper must be changed to include the specific terms for the materials used instead of generic Natural and Synthetic. Further the word composite must be used for clarification.
Response: Thank you for the kind reminders. Your suggestion makes the details of our article easier to understand. We have changed the title to “3D Printed GelMA/PEGDA/F127DA Scaffolds for Bone Regeneration”. If there are suggestions for the revision of the title, you can inform us to make changes.
- The abstract is very generic and must include the material compositions used in the paper.
Response: Thank you for your thoughtful comments. Your opinion is very important to improve the quality of our manuscript. We have added the specific composition of the material to the abstract, and we have also added additional quantitative results to make it easier to understand.
- There are grammar errors throughout the paper and it must be corrected. For example, Pg 1, line 23 the sentence structure must be considered for a change to reflect the intended meaning.
Response: Thank you for your comment. We apologize for the error, which was an oversight on our part, and we have corrected it in the manuscript. Moreover, we invited two native English-speaking experts to revise the manuscript. We will use English language editing by the MDPI if you are still not satisfied with our revised manuscript.
- The unit nomenclature must be consistent across the paper, some of them have space after and some don’t. For example, I have noticed 100 µm and 6 mm etc.
Response: Thank you for your thoughtful comments. We apologize for the errors. The unit nomenclature has been consistent across the paper, such as 100 µm and 6 mm.
- The superscript and subscripts must be checked for all applicable terms as they are not in correct form. For example, CO2, cm2/ml.
Response: Thank you for your thoughtful comments. We apologize for the errors. We checked the manuscript again, and the superscript and subscripts have been in correct form for all applicable terms.
- The paper does not provide justification for choosing the pore sizes of 600 µm. the pore sizes are large compared to the previously established pore sizes in literature yet have not explained why they chose them.
Response: Thank you for your thoughtful comments. Your opinion is very important to improve the quality of our manuscript. An extensive literature search was conducted and a detailed discussion of why 600 µm was used in the Discussion section. Briefly, the pore size can be chosen between 400 and 700 µm, and 600 µm is one of the best results in the literature.
- Appropriate abbreviation expanded terms must be provided to enrich the reader experience for example, CCK-8 and ALP were never abbreviated.
Response: Thank you for your thoughtful comments. We checked the full text and explained in detail when the abbreviation first appeared.
- The LIVE/DEAD assay’s brand and type must be provided.
Response: Thank you for your thoughtful comments. The LIVE/DEAD assay’s brand and type have been provided in line 76.
- The cell seeding procedure must be detailed to included how they were seeded and the apparatus that were used.
Response: Thank you for your comments. Sterile scaffolds were soaked in medium for 15 min in 24-well plates, which were then removed from the medium, and 1 ml of cell suspension (5 × 104 cells) was added. MC3T3-E1 cells were seeded on different scaffolds and cultured in medium for 48 h. We have added the details to the Materials and Methods section in line 147.
- The cell concentrations used in the study must be justified with statements in discussion. Additionally, the 105 cells used in the PCR study, how did the authors come up with the number.
Response: Thank you for your thoughtful comments. We apologize for this, and the cell count should be 105.
- The authors have never provided an image of what a DLP printed scaffold looked like, so it is very important to prove that the structure of the scaffold was not morphologically compromised.
Response: Thank you for your thoughtful comments. Your opinion is very important to improve the quality of our manuscript. We added an image of a DLP-printed scaffold in Figure 2.
- The dimensions of the printed pore and strand size must be provided.
Response: Thank you for your thoughtful comments. We added an SEM image of a DLP-printed scaffold in Figure 2. As predesigned, the printed scaffolds possessed a favorable porous structure, with a pore size of 508.13 ± 21.28 μm for the GPF scaffold, which was lower than 564.04 ± 17.56 μm for the PF and 600 μm for the predesigned scaffold. The pore size of the printed scaffolds was calculated using ImageJ software (NIH, V1.8.0, USA). Three images were selected for each sample, and five pores were measured for each image.
- The quality of the small graphs within the figures must be improved. The graphs pixelate at higher magnifications and at lower magnifications they are blurry.
Response: Thank you for your thoughtful comments. Your opinion is very important to improve the quality of our manuscript. We replaced the images with higher resolution.
- In the discussion section when the references were cited after et al., they must still be positioned at the end of the sentence.
Response: Thank you for your thoughtful comments. Your opinion is very important to improve the quality of our manuscript. The references have been positioned at the end of the sentence.
Reviewer 3 Report
1. The abstract should be broadened to give additional quantitative results.
2. Please end your abstract with a "take-home" message.
3. Rearrange keywords alphabetically.
4. Novelty in the current study's is too weak. The past has seen an extensive study of a lot of written material. It is required to provide more details for more explanation about the present novel in the introductory section.
5. Previous study related needs to explain in the introduction section consisting of their work, their novelty, and their limitations to show the research gaps that intend to be filled in the present study.
6. Why the present article using polymer bone scaffold materials? Not using other materials such as metals? Please explain it rationalization. It is important point that needs to be included by authors in the introduction and/or discussion section. Also, to support this explanation, the suggested reverence published by MDPI should be adopted as follows: Level of Activity Changes Increases the Fatigue Life of the Porous Magnesium Scaffold, as Observed in Dynamic Immersion Tests, over Time. Sustainability 2023, 15, 823. https://doi.org/10.3390/su15010823
7. Rather than relying just on the predominate text as it already exists, the authors could incorporate more illustrations as figures in the materials and methods section that illustrate the workflow of the current study.
8. It is required to include additional information on tools, such as the manufacturer, the country, and the specification.
9. The revised manuscript after peer review must provide detailed information on the error and tolerance of the experimental equipment utilized in this study. Due to the disparate outcomes of other researchers' subsequent studies, it would make for a valuable discussion.
10. Findings must be compared to similar past research.
11. The discussion in present article is extremely poor in quality as overall. The authors must elaborate on their arguments and provide a thorough justification. Don't just state the results and give a quick explanation.
12. Please include the limitation of the present study, it is missing.
13. Line 346-354, in conclusion but not in the conclusion section? It is wired, please revise it.
14. In the conclusion, please explain the further research.
15. The authors need to enrich the reference from five years back. MDPI reference is strongly recommended.
16. The manuscript needs to be proofread by the authors since it has grammatical and language issues.
17. It is suggested to the authors for providing graphical abstract in the system after revision.
Author Response
Dear Reviewer,
Thank you for your insightful advice on our manuscript entitled “DLP-Printed Scaffold Based on Natural-synthetic Polymers for Bone Regeneration” (ID: jfb-2158500). Accordingly, we have revised the manuscript and all amendments are marked up using the “Track Changes” function. In addition, point-by-point responses are listed below this letter.
We hope that the revision is acceptable for publication.
Yours sincerely,
Jianheng Liu
January 25, 2023
Comments:
- The abstract should be broadened to give additional quantitative results.
Response: Thank you for the kind reminders. Your suggestion makes the details of our article easier to understand. We have added the specific composition of the material to the abstract, and we have also added additional quantitative results to make it easier to understand.
- Please end your abstract with a "take-home" message.
Response: Thank you for your thoughtful comments. Your opinion is very important to improve the quality of our manuscript. We have changed the end of the abstract to “The GPF scaffold prepared by DLP printing technology provides a new approach to the treatment of bone defects”.
- Rearrange keywords alphabetically.
Response: Thank you for your comment. We have rearranged keywords alphabetically.
- Novelty in the current study's is too weak. The past has seen extensive study of a great deal of written material. It is required to provide more details for more explanation about the present novel in the introductory section.
Response: Thank you for your thoughtful comments. Your opinion is very important to improve the quality of our manuscript. We have added a brief summary of the material in the Introduction section and suggested shortcomings that still need improvement. Moreover, we specifically discuss this in detail in the Discussion section.
- Previous study related needs to explain in the introduction section consisting of their work, their novelty, and their limitations to show the research gaps that intend to be filled in the present study.
Response: Thank you for your thoughtful comments. Your opinion is very important to improve the quality of our manuscript. PEGDA can be mixed with materials such as nanohydroxyapatite, nanoclay, and extracellular matrix to form bioinks for the preparation of BTE scaffolds. However, the mechanical properties, printability and bioactivity of the scaffolds still need to be improved. We have added a brief summary in the Introduction section and discussed it in detail in the Discussion section.
- Why the present article using polymer bone scaffold materials? Not using other materials such as metals? Please explain it rationalization. It is important point that needs to be included by authors in the introduction and/or discussion section. Additionally, to support this explanation, the suggested reverence published by MDPI should be adopted as follows: Level of Activity Changes Increase the Fatigue Life of the Porous Magnesium Scaffold, as Observed in Dynamic Immersion Tests, over Time. Sustainability 2023, 15, 823. https://doi.org/10.3390/su15010823.
Response: Thank you for your thoughtful comments. Based on your comments, we have discussed this in detail in the Discussion section. Among various materials, natural materials with good bioactivity have disad-vantages such as poor mechanical properties and rapid degradation; inorganic ceramic and some metallic materials (titanium, steel) with strong mechanical properties possess slow degradation and are generally brittle; and some metals with weak rigidity, such as magnesium metal, can promote vascular and bone regeneration, but some studies have demonstrated that the dynamic environment in vivo can lead to accelerated fatigue of magnesium materials, making the magnesium scaffold lose its role earlier. In addition, high-temperature sintering during the preparation of scaffolds using inorganic materials can lead to the inactivation of active substances in bioink, and these disadvantages limit the application of the materials in BTE.
- Rather than relying just on the predominate text as it already exists, the authors could incorporate more illustrations as figures in the materials and methods section that illustrate the workflow of the current study.
Response: Thank you for your thoughtful comments. Your opinion is very important to improve the quality of our manuscript. We have added a schematic diagram of the material preparation and experimental flow for clarity.
- It is required to include additional information on tools, such as the manufacturer, the country, and the specification.
Response: Thank you for your thoughtful comments. Your opinion is very important to improve the quality of our manuscript. We have added additional information to the tools used during the experiment.
- The revised manuscript after peer review must provide detailed information on the error and tolerance of the experimental equipment utilized in this study. Due to the disparate outcomes of other researchers' subsequent studies, it would make for a valuable discussion.
Response: Thank you for your thoughtful comments. We detail the possible errors in the discussion section. For example, the accuracy of the universal tensile machine is 0.5%, which means that the mechanical results may have an error of 0.5%.
- Findings must be compared to similar past research.
Response: Thank you for your thoughtful comments. Your opinion is very important to improve the quality of our manuscript. Some scholars have conducted similar studies in the past, and we compare them in the discussion section, such as increased bioadhesive properties and mechanical properties over similar materials.
- The discussion in present article is extremely poor in quality as overall. The authors must elaborate on their arguments and provide a thorough justification. Do not just state the results and give a quick explanation.
Response: Thank you for your thoughtful comments. Your opinion is very important to improve the quality of our manuscript. In the discussion section, we added the reasons for the material selection and the pore size of 600 μm; moreover, in comparison with the previous materials, we listed our advantages and shortcomings.
- Please include the limitation of the present study, it is missing.
Response: Thank you for your thoughtful comments. Your opinion is very important to improve the quality of our manuscript. The following limitations may exist. First, the accuracy of the universal tensile ma-chine is 0.5%, which means that the mechanical results may have an error of 0.5%. Second, the differential outcomes of GPF scaffolds on osteogenic differentiation in diverse environments still require further investigation. We added this information to the end of the Discussion section.
- Line 346-354, in conclusion but not in the conclusion section? It is wired, please revise it.
Response: Thank you for your thoughtful comments. Your opinion is very important to improve the quality of our manuscript.
- In the conclusion, please explain the further research.
Response: Thank you for your thoughtful comments. We have initially combined the Conclusion with the Discussion section. As you suggested, we have included the Conclusion section separately.
- The authors need to enrich the reference from five years back. MDPI reference is strongly recommended.
Response: Thank you for your thoughtful comments. Your opinion is very important to improve the quality of our manuscript. In revising the Introduction and Discussion, we reviewed the literature from 5 years back and added them to the references.
- The manuscript needs to be proofread by the authors since it has grammatical and language issues.
Response: Thank you for your thoughtful comments. Your opinion is very important to improve the quality of our manuscript. We invited two native English-speaking experts to revise the manuscript. We will use English language editing by the MDPI if you are still not satisfied with our revised manuscript.
- It is suggested to the authors for providing graphical abstract in the system after revision.
Response: Thank you for your thoughtful comments. Your opinion is very important to improve the quality of our manuscript. We have provided a graphical abstract in the system as you suggested.
Reviewer 4 Report
Dear Authors,
The manuscript is well written and it is recommended to accept in its present form.
Thanking you,
Author Response
Dear Reviewer,
Thank you for your insightful advice on our manuscript entitled “DLP-Printed Scaffold Based on Natural-synthetic Polymers for Bone Regeneration” (ID: jfb-2158500). Accordingly, we have revised the manuscript and all amendments are marked up using the “Track Changes” function. In addition, point-by-point responses are listed below this letter.
We hope that the revision is acceptable for publication.
Yours sincerely,
Jianheng Liu
January 25, 2023
Comments:
The manuscript is well written and it is recommended to accept in its present form.
Response: Thank you for your recognition of our manuscript. We will continue to work hard.
Round 2
Reviewer 3 Report
Well done to the authors, I have some other comments as my response in the revised version:
1. Why the tittle change? Any specific reasons?
2. Line 45, the authors write “Poly (ethylene glycol)”, please revise it as “Polyethylene glycol”
3. Please explain the utilization of computational simulation for further research that would be advantages compared to experimental testing as compared in present article, such as lower cost and faster results. Also, additional relevant reference is needed as follows: The Effect of Bottom Profile Dimples on the Femoral Head on Wear in Metal-on-Metal Total Hip Arthroplasty. J. Funct. Biomater. 2021, 12, 38. https://doi.org/10.3390/jfb12020038
Author Response
Dear Reviewer,
Thank you for your insightful advice on our manuscript entitled “3D Printed GelMA/PEGDA/F127DA Scaffolds for Bone Regeneration” (ID: jfb-2158500). Accordingly, we have revised the manuscript and all amendments are marked up using the “Track Changes” function. In addition, point-by-point responses are listed below this letter.
We hope that the revision is acceptable for publication.
Yours sincerely,
Jianheng Liu
January 28, 2023
Comments:
- Why the tittle change? Any specific reasons?.
Response: Thank you for the kind reminders. We changed the title because one of the reviewers suggested that the title of the paper must be changed to include the specific terms for the materials used instead of generic Natural and Synthetic. For the reason as above, we changed the title.
- Line 45, the authors write “Poly (ethylene glycol)”, please revise it as “Polyethylene glycol”
Response: Thank you for your thoughtful comments. We apologize for the errors and have corrected it in the revised manuscript.
- Please explain the utilization of computational simulation for further research that would be advantages compared to experimental testing as compared in present article, such as lower cost and faster results. Also, additional relevant reference is needed as follows: The Effect of Bottom Profile Dimples on the Femoral Head on Wear in Metal-on-Metal Total Hip Arthroplasty. J. Funct. Biomater. 2021, 12, 38. https://doi.org/10.3390/jfb12020038
Response: Thank you for your comment. Just like you said, the establishment of a 3D finite element model to simulate 3D physiological loading has made significant progress in the design of implants, which has given us great insight to predict the state of BTE scaffolds in vivo through computational simulations for developing optimal structure. We have added this section to the Discussion and added relevant references.